Ten quick tips for clinical electroencephalographic (EEG) data acquisition and signal processing

Cisotto Giulia 1 2
Chicco Davide davide.chicco@gmail.com 1 3
1 Dipartimento di Informatica Sistemistica e Comunicazione, Università di Milano-Bicocca , Milan , Milan , Italy
2 Dipartimento di Ingegneria dell’Informazione, Università di Padova , Padua , Padua , Italy
3 Institute of Health Policy Management and Evaluation, University of Toronto , Toronto , Ontario , Canada
Chen Vincent
Electronic publication date: 2024 Sep 3
Publication date: 2024
Volume: 10
Electronic Location ID: e2256
Received 2024 May 23; Accepted 2024 Jul 22
Copyright: ©2024 Cisotto and Chicco
Copyright year: 2024
Copyright holder: Cisotto and Chicco
License: This is an open access article distributed under the terms of the Creative Commons Attribution License, which permits unrestricted use, distribution, reproduction and adaptation in any medium and for any purpose provided that it is properly attributed. For attribution, the original author(s), title, publication source (PeerJ Computer Science) and either DOI or URL of the article must be cited.
License URL: https://creativecommons.org/licenses/by/4.0/

Keywords: EEG, Electroencephalography, Quick tips, Signal processing, Medical signal processing

Funding: European Union–Next Generation EU programme, in the context of the National Recovery and Resilience Plan, Investment Partenariato Esteso PE8 “Conseguenze e sfide dell’invecchiamento”, Project Age-It (Ageing Well in an Ageing Society) Ministero dell’Università e della Ricerca of Italy to Università di Milano-Bicocca (Milan, Italy) Dipartimenti di Eccellenza 2023-2027 Dipartimento di Informatica Sistemistica e Comunicazione at Università di Milano-Bicocca The work of Davide Chicco is funded by the European Union–Next Generation EU programme, in the context of the National Recovery and Resilience Plan, Investment Partenariato Esteso PE8 “Conseguenze e sfide dell’invecchiamento”, Project Age-It (Ageing Well in an Ageing Society). Giulia Cisotto is also supported by the financial support of PON “Green and Innovation” 2014-2020 action IV.6 funded by Ministero dell’Università e della Ricerca of Italy to Università di Milano-Bicocca (Milan, Italy). Giulia Cisotto and Davide Chicco are supported by Ministero dell’Università e della Ricerca of Italy under the “Dipartimenti di Eccellenza 2023-2027” ReGAInS grant assigned to Dipartimento di Informatica Sistemistica e Comunicazione at Università di Milano-Bicocca. There was no additional external funding received for this study, The funders had no role in study design, data collection and analysis, decision to publish, or preparation of the manuscript.

==============================
Electroencephalography (EEG) is a medical engineering technique aimed at recording the electric activity of the human brain. Brain signals derived from an EEG device can be processed and analyzed through computers by using digital signal processing, computational statistics, and machine learning techniques, that can lead to scientifically-relevant results and outcomes about how the brain works. In the last decades, the spread of EEG devices and the higher availability of EEG data, of computational resources, and of software packages for electroencephalography analysis has made EEG signal processing easier and faster to perform for any researcher worldwide. This increased ease to carry out computational analyses of EEG data, however, has made it easier to make mistakes, as well. And these mistakes, if unnoticed or treated wrongly, can in turn lead to wrong results or misleading outcomes, with worrisome consequences for patients and for the advancements of the knowledge about human brain. To tackle this problem, we present here our ten quick tips to perform electroencephalography signal processing analyses avoiding common mistakes: a short list of guidelines designed for beginners on what to do, how to do it, and what not to do when analyzing EEG data with a computer. We believe that following our quick recommendations can lead to better, more reliable and more robust results and outcome in clinical neuroscientific research.

Introduction

Electroencephalography (EEG) is a convenient and common tool to record the electrical activity of our brain in a non-invasive way, easily, at relatively low cost, and eventually using portable devices. The EEG allows to accurately follow the fast dynamic of the brain and to obtain quantitative measurements of the electrical activity that small portions of our brain produce while we are accomplishing cognitive and motor tasks, as well as while we are in resting-state. The electroencephalogram is a special case of electrogram, and should not be confused with the electrocardiogram (ECG), which is the recording of the electrical activity of the heart.

The EEG differs from other kinds of brain monitoring, for example functional magnetic resonance imaging (fMRI) or magnetoencephalography, to need cheaper costs, lighter and simpler acquisition procedures, faster acquisition sessions, and to be portable. For these reasons, EEG is one of the most popular methods to investigate the brain and tons of algorithms have already been proposed to analyze this kind of signal.

The first quantitative analysis of an EEG signal dates back to the pioneering work of Hans Berger who, in 1929, employed a Fourier transform of an EEG signal to quantify the spectral distribution of the brain activity under different physiological and stimulation conditions (Berger, 1929). Since then, a vast literature flourished and obtained successful achievements in the modeling and classification of EEG data for different clinical and research applications (Teplan, 2002). However, the use of electroencephalography presents several challenges, with non-trivial choices to design and implement the acquisition steps, and a critical, yet underestimated, pre-processing phase.

Researchers take advantage of EEG data processing in several scientific fields, and not only in medicine: EGG data in fact are the core of analyses on emotion recognition (Xu, Guo & Wang, 2022), sleep stage classification (Zaman et al., 2024), motor imagery classification (Zaman et al., 2024; Zancanaro et al., 2021), neurorehabilitation (Ang & Guan, 2016), seizure detection (Shen et al., 2024), just to mention a few.

Several articles serve as guides to beginners on how to understand the electroencephalography (EEG). The studies of Beniczky & Schomer (2020) and Biasiucci, Franceschiello & Murray (2019), for example, describe the basic features of the EEG, how its data are encoded and how they can be analyzed.

Some studies published in the past described the main challenges and obstacles of EEG signal computational processing (Khademi, Ebrahimi & Kordy, 2023; Sharma et al., 2023; Rashid et al., 2020; Sun & Mou, 2023; Ein Shoka et al., 2023). These research works capture efficiently the main issues of the EEG computational signal processing scenario, indicating problems that we ourselves describe in the present article as well. However, the authors of these studies do not provide practical solutions for handling these obstacles: we tackle this problem by providing here our ten simple tips to perform EEG signal processing correctly.

So far no article in the scientific literature explains how to perform EEG signal processing correctly avoiding common mistakes: we fill this gap by presenting here our guidelines for this common digital signal processing activity. Our ten simple recommendations, if taken into account, can help researchers avoid common mistakes and perform better, more robust analyses that, in turn, can lead to more reliable results and outcomes (Fig. 1). Even if we wrote these guidelines for beginners, we believe they should be followed by experts, too, in any scientific project involving EEG data.

Figure 1 Schematization of the proposed tips as a flowchart.

This diagram shows all the steps we suggest to follow in order to facilitate the integration and analysis of electroencephalografic data. We generated the icons through Microsoft Windows PowerPoint.

Tip 1: Before starting, clearly define your scientific question and your experimental protocol with a clinical neuroscientist

The first action to take when you have a research topic in mind is to clearly define a research question. In this field, where multidisciplinarity is an essential characteristic of the investigation, you must discuss and converge towards a consistent and well-grounded research question together with the clinicians and the other members of the research team (it might include neurologists, neuroscientists, physical therapists, cognitive psychologists, data scientists, etc.).

The research question must be well-grounded from the current state of the art, innovative, simple to be explained, and clearly stated (Fig. 1).

Once it is formulated, a research protocol has to be properly designed. Every in-field experiment has to be supported by a rigorous protocol that defines the methods and the timing of the data collection. The protocol typically includes:

1. A systematic review of the state-of-the-art;

2. The research question and the main objective of the investigation as well as intermediate objectives;

3. The methods to reach the objectives, including the type of study to conduct (for example, a randomized clinical trial), the equipment to collect the data, the statistical analysis and the numerical computation planned to be used, the reporting strategies, and the validation methods;

4. The criteria to include and exclude participants from the data collection and their assignment to the experimental groups (the “control group” typically includes healthy subjects, while the “treatment group” is composed by patients receiving a specific therapeutic intervention);

5. The dissemination and publication modalities at the end of the investigation.

Research protocols ensure the robustness of the investigation (that is, of its outputs), the safety of the participants, and the replicability of the study all over the world.

You can find templates open to be used as a basis to design your own research protocol in Higgins & Green (2008) (particularly, its Box 2.2a) and in The Cochrane Collaboration (2024).

If working at the hospital, the research protocol has to be approved before it can be implemented, any research involving humans must be approved by an Institutional Review Board (IRB), and informed consent from the subjects must be collected before participation. Then, it is mandatory that it is presented to the hospital’s Ethical Committee, a multidisciplinary committee formed by clinicians, statisticians, experts of medical law and insurance, professionals in health-tech (medical equipment and devices) and possibly other stakeholder representatives. The committee has the role of analyzing all parts of the submitted protocol and guaranteeing its applicability and safety for the involved participants (healthy and patients, as well).

Once you have your experimental protocol approved, you and your team can start recruit patients and collect data (Fig. 1).

Tip 2: Keep in mind that part of the success of your scientific project depends on the quality of the EEG recordings

To ensure proper medical interpretation of the raw EEG data as well as the success of any machine learning model (ML), it is recommended to make the best effort to obtain high-quality EEG recordings (Fig. 1). This can be realized by following these simple guidelines:

1. If available, use high-quality research-grade equipment. This typically provides finer resolution (in time and amplitude) of the EEG data.

2. Carefully and properly set the acquisition parameters. First, the sampling frequency should be high enough to allow you to detect the brain behaviour under investigation. If brain reaction is expected a few milliseconds after a given stimulus, the sampling frequency should be higher than 1 kiloHertz (kHz), in order to have at least 1 sample every millisecond. Typical values for the sampling frequency are: 500, 1,000, 2,000 Hz. Pay attention also to have a sufficient number of bit to code the values of your signals: bit resolution should be at least 8 bit (check the technical specifications of the EEG device you use). The number of channels is also important: having a sufficiently high number of sensors (also called electrodes) covering the whole scalp surface allows you to monitor the whole brain activity, that is the activity produced by all regions of the brain. On the other hand, if the medical team has strong assumptions on the region where the activity of interest is produced, you can decide to reduce the number of sensors, having a lighter setup with consequent lower time for the preparation of the participant. Finally, there are two special electrodes to place: the ground and the reference one. Our advice is to place the reference electrode in an electrically-neutral location (like the ear lobe or the mastoid). If possible, use the linked-mastoids or linked-ears function provided by the EEG equipment company, to ensure perfect symmetry in the acquisition. For the ground, you can find indication by the company of which electrode has to be used as ground (check the equipment user manual).

3. Follow the International 10–20 EEG Placement System (or its extension) to correctly place the EEG electrodes on the participant’s scalp (Homan, Herman & Purdy, 1987). More often, an EEG cap is provided by the EEG producer which integrates all sensors in agreement with the above-mentioned standard. On the other hand, in case you use a portable device, the electrode placement might be different to improve the user comfort. In this case, the reference to the International 10–20 EEG Placement System is useful during the subsequent analysis to compare your results with other studies using standard equipment. Ensure no hair is in between the EEG electrodes and the scalp, or try to scrap them apart. To note, require the participant not to apply any hair foam before the EEG experiment. Bald people might show lower signal because of a thicker scalp. Ensure that cables from the cap to the amplifier are not stretched anywhere.

4. Ensure that the participant sits comfortably on the chair, with the monitor (if any) at about 1 meter apart from them at their eyes’ height (so to avoid head movements).

5. To ensure the fairest comparison across different EEG studies, make sure that participants are recorded at the same time during the day (to have consistent phase of the circadian rhythm) and that the room is properly shielded from other electromagnetic interferences.

6. During montage, fill the space between each electrode and the scalp with enough conductive gel to have a sufficiently low impedance that allows a good signal amplitude. You can read the impedance values in the software provided with the equipment. Check the value of every single channel, starting from the ground and the reference ones. Remember: the amplifier takes the voltage difference between each electrode and the reference one, and calibrates itself on the electrical activity acquired at the ground location. This means that if the reference and/or the ground electrodes have problems, all the remaining electrodes will suffer from poor quality. The rule-of-thumb is to keep the impedance value below 20 kiloOhm (but check the equipment user manual, because this threshold might change depending on the electronics of each specific EEG amplifier).

7. Once step 5 is completed, check the raw signals on the visualization panel (equipment software). The signals (from any channel) is expected to have small variations in the range of (max) ±100 microvolts (µV). Figure 2 represents an example of multi-channel EEG recording. If you notice large and slow variations, you should go back to step 6 and check the impedance of the reference, the ground, and the electrodes displaying problems.

8. Before starting the actual data recording, ask the participant to blink their eyes repeatedly, to clench teeth, and, finally, to close eyes and relax. You should verify, respectively, that frontal electrodes show a rhythmic activity corresponding with the blinking rhythm, that temporal and fronto-temporal electrodes show electromyography (EMG) activity overlapping the EEG signal while the participant is clenching, and finally that a small periodic activity (at about 10 Hz, the famous α rhythm) appears at the parietal and occipital electrodes while the participant is fully relaxing. Figure 3 shows an ideal example of clean EEG recording, with the clear occurrence of some α rhythm events.

9. Finally, instruct the participant to stay as much relaxed as possible during the entire experiment duration. Remember, every movement (for example, head, legs, etc.) might induce artefactual components on the EEG recordings. If the experiment takes long, tell them to ask for a break whenever they feel cognitive fatigue or drowsiness.

Tip 3: Properly clean your data before starting the computational analysis: check time course and frequency spectrum, and remove noise

Before starting any computational step, we strongly suggest you check the raw data. You need to plot the signals both in the time and in the frequency domain. In the time domain, check that the amplitudes do not exceed ±100 µV (well-established rule of thumb). In the frequency domain, you should recognize the 1/f shape of the power spectrum. On the top of the 1/f shape, you might recognize some additional narrowband peaks around 5 Hz, 10 Hz and 20 Hz, corresponding to the clinical theta, alpha and beta bands. These frequency bands have a important clinical value. Remember that EEG cannot reliably capture brain frequencies above 90 Hz. Then, if you find large peaks above that frequency, it is much likely that they convey non-brain information and should be eliminated (Fig. 1).

Figure 2 An example of EEG multi-channel recordings.

Image released on Wikimedia Commons under the Creative Commons CC0 1.0 Universal Public Domain Dedication (Krol, 2024). Each signal represents one EEG recording from one single channel (channel’s name on y axis) along time (x axis).

Figure 3 An example of clean EEG recordings with visible α ryhthm events (in the red circles).

To remove the unwanted components, it is common to use filters. The most general filters you can apply are: a notch filter to remove the power line noise (50 or 60 Hz, depending on the country where you work). Then, you need to consult the literature or your colleagues from the clinical side, to decide other filters to apply. To make a few examples, in the case you are studying brain processing in relation to a cognitive task or sensory task, you might heavily filter the signal below 30 Hz (for example, to detect the popular P300 component (Polich, 1993)). On the contrary, when dealing with movements and desynchronization of sensorimotor rhythms (Pfurtscheller & Da Silva, 1999), we suggest you to apply a band-pass filtering between 3 and 45 Hz (or 55 Hz, in case power line is at 60 Hz).

To increase the spatial specificity of the signal acquired from one location, you can also apply spatial filters such as the common average (CAR) filter, the small Laplacian filter, or the large Laplacian filter. We recommend not to use the CAR, unless you are sure to have clean signals from all electrodes. Spatial filtering is more beneficial in high-density recordings (more than 64 channels), where the electrodes spacing is short.

In the latter cases, you might also decide to exclude the signal from a particularly corrupted channel and substitute it with the interpolated version of its surrounding channels. The latter contain similar information to the rejected channel, given their short distance from it.

Finally, it is suggested to run independent component analysis (ICA) (Makeig et al., 1995) to decompose the multi-channel recording into independent components, identify those most likely associated with artefacts (for example, eye blinks), and then recompose the signal by using only clean components. To perform this procedure, the help of domain experts or other online resources (for example Pion-Tonachini, Makeig & Kreutz-Delgado, 2024; Pion-Tonachini, Makeig & Kreutz-Delgado, 2017) is needed to properly select which components to discard. With this method, you can reliably remove the power line noise, blinks and, to some extent, also physiological interferences (for example, muscular or heart activity). Once you obtain the new multi-channel EEG recording after ICA, check the quality of the new dataset.

As a side suggestion, keep track of the signals or the components you discard. Pre-processing in EEG is often not fully replicable, so having notes of what pieces of the signal have been eliminated and why might be important in the case of article review or to reuse the same dataset in the future (Tip 10) (Karimzadeh & Hoffman, 2018; Vardigan, Heus & Thomas, 2008; Rasmussen & Blank, 2007; Fabris et al., 2022).

Tip 4: Always pay attention to time, frequency, and space domains, subject-specificity, and inter-subject variability of the EEG signals

Assuming you have a clean EEG dataset, it is of key importance to analyze it in the three main domains: time, frequency, and space (Fig. 1). The time domain is essential to identify specific brain responses to stimuli that might be included in the experimental protocol (a typical example is the P300, explained in Tip 3). The frequency domain is needed especially when you expect a particular engagement of specific brain waves. Common examples are the steady-state visual evoked potentials (SSVEP) or the event-related desynchronization (ERD) paradigms: in SSVEP, a visual stimulus is repeatedly administered to the subjects letting their brain synchronize the occipital activity to the same frequency of the stimulation (Beverina et al., 2003). In the ERD paradigm (task-oriented), while the subject is moving, movement-related brain areas decrease their activity in two frequency bands around 10 Hz and 20 Hz. In these cases, comparing the power spectra before and during stimulation/task performance will make these frequency responses clearly appearing. The space domain is fundamental to find the regions where the above time and frequency responses are more evident. As an example, ERD is mostly found over the sensorimotor cortex (for example, C3, Cz, C4 electrodes) (Dugué et al., 2020), SSVEP over the parietal and occipital cortex (Ding, Sperling & Srinivasan, 2006), while P300 is mostly visible over the centro-parietal midline (Cz, Pz).

To optimize the analysis efforts towards the most informative direction, it is critical to have prior knowledge or hypotheses coming from the clinical side or neuroscientific literature. Depending on the task performed by the individual (motor imagery, sleep, rest with open eyes, etc.), the brain electrical activity can highly vary. Also, keep in mind that other two factors can change the shape of the EEG recordings: even using the same experimental protocol (Fig. 1), that is, asking the participants to perform the same task with the same timing, you might see differences in their EEG signals due to possible pathology affecting them, and also due to the inherent inter-subject variability that characterizes the human brain.

In the analysis of EEG data, identifying common patterns across different subjects and investigating subject-specific responses are both important.

Tip 5: Properly prepare your dataset for the computational analysis

To identify those time-frequency-space patterns associated with a task or a pathological brain behaviour, machine learning (ML) models are typically employed. However, it is common to run a few pre-processing steps to prepare the clean dataset for being input to the model (Fig. 1). First, segmentation is performed. This means to cut the long-time EEG recording, into smaller chunks that will be then independently treated. Choosing the correct segment duration is a non-trivial step. We would suggest to choose the duration in relation to the experimental protocol: for example, if a movement task is asked to be repeatedly performed every 10 seconds, then a good option is to cut the EEG signals to have 10s-segments. On the other hand, if no external time scheduling is imposed to the subject (in sleep or resting state studies), you might consider 2-seconds segments. The latter is an empirical numerical choice representing a popular choice in the EEG community, as it should allow to ensure stationarity (that is, the same statistical properties) across different segments and, thus, proper analysis. Besides, we strongly recommend you to segment after filtering, to avoid filter border effects to appear in every segment, attenuating a good amount of initial and final segment samples.

Second, normalization can be applied if you need to level out differences in those factors that you don’t want to influence your analysis. For example, if you are interested in the general brain response to a specific new stimulus without considering individual differences, then normalizing EEG signals within each subject can be a good choice.

Lastly, for most ML models you need to extract a number of well-established features: a rich list of possible features can be found in Cisotto et al. (2020b). Features can be extracted from every single electrode, both in the time and in the frequency domain. Then, feature selection based on prior expert knowledge or using an automatic algorithm (Cisotto et al., 2022; Cisotto et al., 2020a) might be beneficial to let the machine learning model learn more effectively and quickly. Once you have transformed your raw EEG dataset into the features domain, then normalization of each single feature (across EEG samples) might be useful to help the model learn.

If you use deep learning techniques, then feature extraction could be skipped, and you can decide to input raw segments of EEG data into the model: however, this choice depends on the specific architecture you use. There is no golden choice, you need to carefully study previous literature addressing similar analysis.

However, if your objective is to prove the effectiveness of a new analytical method, then use the same input size and preprocessing steps before compare your results with previous literature results.

Tip 6: Carefully choose between the “sensors” or a “sources” approach

When talking about EEG analysis, you can decide to work with the data directly coming from the electrodes (sensor analysis), or to infer the brain activity of deeper brain regions using specific models to transform the recorded EEG data into the real brain sources (source approach) (Fig. 1). The latter operation lets you find the most probable sources of the electrical activity you have just measured at the surface of the head.

In the following, we explain how to carry source analysis and we discuss the opportunity and the disadvantages to evaluate when deciding to work with sources or sensors. This tip has to be a bit more technical to allow you understand the complexity, but also the advantages, of the source approach.

In some clinical applications, it might be useful to operate the so-called electrical source imaging (He et al., 2011; Michel & He, 2019), a transformation aimed to map the electrical activity measured at the scalp level (by the EEG sensors) to a set of brain sources that might have produced those superficial measurements. This processing is particularly useful in clinical applications like the identification of the sources of epileptic foci (Bénar et al., 2006), of the α rhythm activity during a resting-state period (Cuspineda et al., 2009), of the sleep waves (spindles) (Del Felice et al., 2014), and in examinations where other imaging methods (MRI, fMRI, CT, etc.) are available and can be used in co-registration with EEG (Bénar et al., 2006).

However, implementing this transformation is a non-trivial task. There is an entire subfield of computational neuroscience addressing the challenges related to this transformation: in fact, a number of hypotheses has to be assumed. First, you need to have a model of the head layers (Plummer, Harvey & Cook, 2008). Second, prior knowledge should let you select the number and the distribution of the sources expected to explain the superficial activity (He et al., 2011; Michel & He, 2019). Third, a variety of possible models to actually realize the mapping are available: to name a few, the weighted minimum norm (WMN) (de Peralta Menendez et al., 2004) , the popular low-resolution electromagnetic tomography algorithm (LORETA) (Pascual-Marqui, Michel & Lehmann, 1994), the multiple signal classification (MUSIC) (Mosher & Leahy, 1999). The interested reader might refer to Kaur et al. (2022), Asadzadeh et al. (2020) for recent reviews on this topic.

Nevertheless, if you have less than 60 electrodes, it is not recommended to apply this transformation: motivated by the fundamental theorems of linear algebra and mathematical modeling, you need to have a sufficiently high number of sensors to reverse the modelling problem and obtain the sources of the observed activity. Empirically, 60 sensors is a well-accepted minimum value to have a reliable source localization within the brain from scalp EEG recordings (Michel et al., 2004).

On the contrary, if you have more than 60 electrodes and you opt for the source approach, you can find a number of software tools and libraries to help you apply this transformation, implementing different modeling methods. To name a few: CARTOOL (Michel & Brunet, 2019), EEGLAB (Delorme et al., 2011), and FieldTrip (Oostenveld et al., 2011).

Finally, it is worth to mention that models based on deep learning have been also recently proposed, showing promising results (Liang et al., 2023; Cui et al., 2019; Pantazis & Adler, 2021).

Tip 7: For the computational analysis, start with the simplest methods, and use more complex methods only if it is necessary

When the dataset you would like to analyze is finally pre-processed and ready to be used, you come to an important decision which regards all the computational researchers in any field: which computational method should I start with? We have a straight piece of advice for this choice: start with the simplest method available. Period.

A simple technique (for example, linear regression), in fact, gives you the possibility to understand how its statistical model works, to comprehend its functioning, and to interpret its results and why they were generated that way. Complex methods, instead, are difficult to implement and hard to interpret, and should be utilized only if necessary. If you obtain sufficient results with a simple method, stick with them; otherwise, of course feel free to use more complex methods. But do not start with complicated methods: start with simple algorithms.

If you can achieve sufficiently relevant results with basic statistics tools (for example, mean, standard deviation, median, minimum and maximum) or with traditional biostatistics tests (such as Mann–Whitney U test (MacFarland et al., 2016), Student’s t test (Mishra et al., 2019), Kruskal–Wallis test (Ostertagova, Ostertag & Kováč, 2014), chi-squared test (McHugh, 2013, etc.), then go with them. For probability values, we suggest to use the p < 0.005 significance threshold, as indicated by Benjamin et al. (2018).

If your analysis involves supervised machine learning, we suggest you to start with Decision Trees, and not to start with complex or large deep learning models (Chicco, 2017; Zancanaro et al., 2021; Cisotto et al., 2023). If your analysis includes an unsupervised machine learning phase, we advise you to start with k-means clustering and not with spectral clustering.

Leonardo Da Vinci used to say: “Simplicity is the ultimate sophistication”. It is true also for EEG signal processing.

Tip 8: Look for a validation cohort dataset online and repeat your analysis on it

Once a researcher has performed a computational analysis on an electroencephalographic dataset and has discovered something relevant regarding the subjects involved, they might tend to think that their job is done and they can focus on paper writing. Actually, scientific findings obtained on a single dataset, although possibly useful and interesting, result being clearly dataset-specific and lack generalizability.

To make a scientific study sounder and more robust, a confirmatory analysis on a validation cohort dataset is necessary (Fig. 1). Finding a dataset that is compatible and similar to the primary cohort dataset employed in a project might be difficult, but fortunately there are multiple online resources in the Internet for this purpose.

So, here is our tip: when you finished your computational analysis on the primary dataset, look for a validation dataset online on resources such as the following ones:

• Google Dataset Search (Google, 2024)

• re3data.org (re3data, 2024)

• PhysioNet (PhysioNet, 2024)

• OpenNeuro.org (Markiewicz et al., 2021)

• EEGLAB Wiki (Delorme et al., 2019; Martínez-Cancino et al., 2021)

• Zenodo (Zenodo, 2024)

• Kaggle (Kaggle, 2024a)

• University of California Irvine Machine Learning Repository (University of California Irvine, 2024)

• Figshare (Figshare, 2024)

• Brain-Computer Interface (BCI) Competition IV (Brain-Computer Interface (BCI) Competition IV Organizers, 2024)

You might also find a suitable dataset among the articles published in the Scientific Data journal (Babayan et al., 2019; Cao et al., 2019; Shin et al., 2018; Luciw, Jarocka & Edin, 2014; Hollenstein et al., 2018; Won et al., 2022; Nieto et al., 2022; Pernet et al., 2019; Grootswagers et al., 2022; Mikulan et al., 2020; Ma et al., 2022; Valdes-Sosa et al., 2021; Stevenson et al., 2019). On Google Scholar, this search can be performed by using the following query: EEG source:“Scientific Data”

The possibility to repeat your computational analysis on an alternative, independent dataset, and perhaps to find similar outcomes to the ones you found on the primary dataset would make your scientific study more reliable and relevant for the scientific community.

Tip 9: Ask a clinical neuroscientist to assess your results

As we mentioned early (Tip 1), a sound medical project always starts with a clear, feasible scientific question designed by clinical neuroscientists. A scientific question is well posed if, once solved, its solution improves the knowledge of neurological research, and consequently can influence medical knowledge and neurological therapies.

So here is our piece of advice. When you generate results on the EEG data you analyzed, we suggest you to knock again on the door of the clinical neuroscientists that you met at the beginning of your journey, and ask them to assess and review your results. They will evaluate your outcomes and findings, providing precious feedback on what to do next, on what to repeat, and on how to write the scientific paper about your electroencephalography project (Fig. 1). A sound medical research project starts in the hospital and ends in the hospital (Fig. 4).

Figure 4 Schematic representation of a sound biomedical research project cycle.

A relevant scientific question is born in a hospital from medical doctors or clinical neuroscientists who identify a current lack or problem in biomedical research or in clinical practice. A scientific question invented by the biomedical engineers or by health informatics researchers, without the help of clinical neuroscientists, is probably badly posed or misleading. Scientific researchers take the medical question in custody from the hospital, then collect the data and pre-process them for the computational analysis. They use computational methods to infer new knowledge on these data, and eventually deliver their scientific results back to the clinical neuroscientists of the hospital where the scientific question was born at the beginning. The clinical neuroscientists review the results, provide feedback, comments, prompts, and insights, and possibly change their strategy on treatments and therapies for patients. The hospital building image was released under the Creative Commons 4.0 BY-NC DEED license on PngAll.com.

In your results delivery document, we advise you to write your report in a simple scientific language that can be understood by clinical neuroscientists, too: avoid jargon, avoid words which have different meanings in different fields, and have a lots of patience when working with medical doctors (Chicco & Jurman, 2023). If possible, ask your clinical colleagues to meet and discuss your results in person.

In this case, put some effort to carefully prepare nice and intuitive figures and plots for the clinicians. Remember to put clear labels, units, and titles on every figure and to describe them in simple words (this can be a useful preparation to have your nice figures for the scientific manuscript you want to write, as well). If possible, arrange the figures in a similar way as done in the papers you use as main references for your study. It will help your colleagues get the meaning and impact of your results.

Our personal experiences showed that, when medical doctors were involved both for the study design and for the results’ assessment, the final results turned out to be sounder and more reliable (Cerono, Melaiu & Chicco, 2023; Chicco et al., 2023).

Tip 10: Follow the principles of open scientific research and document everything

In the history of humanity, scientific progress has happened only when information was shared openly between people. Scientific research, as well, works better when it is open.

We can describe open scientific research through five pillars: usage of open source software code; open software code release; open data release; open access publication; open and complete documentation (for a more detailed, and general, approach to open and reproducible science, also check (Cabitza & Campagner, 2021)).

1. Use only open source programming languages and platforms, such as Python and R. Using an open programming language will allow the sharing of software scripts between collaborators, without any issues regarding licenses. Moreover, it will allow the reproducibility of the experiments, allowing anyone in the world with a computer to install the desired software packages and re-run the tests you did. Python is the most used programming language in the world, according to the PYPL index (PYPL, 2024), to the TIOBE (TIOBE, 2024) and to the Kaggle survey (Kaggle, 2024b). Moreover, the Python package index pypi contains around fifty software libraries for electroencephalography signal processing in stable or mature development status (pypi, 2024). Among them, MNE-Python (Gramfort et al., 2013; Gramfort et al., 2014), Py-EEG (Bao, Liu & Zhang, 2011), and NeuroKit2 (Makowski et al., 2021) are the widely used. R is another open source programming language and platform commonly utilized in health informatics and bioinformatics. A few R software packages for EEG signal processing exist in the Comprehensive R Archive Network (CRAN): eegkit (EEGkit, 2024), MedicalImaging (MedicalImaging, 2024), and eegUtils (EEGUtils, 2024).

2. Release your software code openly online. To enforce the reproducibility of your study, you can also consider publishing your software code openly on GitHub, GitLab, or SourceForge. Moreover, if your software code can be used as a package, you can consider submitting it to a central repository of software libraries, such as pypi for Python, CRAN for R, Julia Packages for Julia or Crates.io for Rust, for example. Additionally, anyone will be able to assess your software scripts and understand if any mistakes were made, making your study methodology more transparent and robust (Barnes, 2010).

3. Release your dataset online, if you are authorized. If the subjects of the EEG experiments gave consent, and the bioethical committee of your institution agreed, go on and publish your EEG data openly online on public data repositories such as PhysioNet (PhysioNet, 2024), OpenNeuro.org (Markiewicz et al., 2021), EEGLAB Wiki (Delorme et al., 2019; Martínez-Cancino et al., 2021), Zenodo (Zenodo, 2024), Kaggle (Kaggle, 2024a), University of California Irvine Machine Learning Repository (University of California Irvine, 2024), or Figshare (Figshare, 2024). Releasing your dataset openly online would make it available for secondary analyses to anyone in the world, who might discover something new regarding clinical neurosciences. Moreover, it would increase the impact of your study. Remember to anonymize the records prior release.

4. Publish your scientific article in an open access journal. If you have a say in what journal you and your team can submit the article about your EEG study, we strongly suggest to pick an open access one. By publishing your article open access, in fact, you would make it available for free to anyone in the world, including to people in the least developed countries, students of high schools, and taxpayers. Additionally, open access articles obtain more citations (Tang, Bever & Yu, 2017). A list of health informatics open access journals where to submit an article on electroencephalographic data can be found on ScimagoJR (Scimago Journal Ranking, 2024). While carefully selecting a journal where to submit a scientific article, it is pivotal to avoid predatory journals (Cobey et al., 2018).

5. Write open and complete documentation. Documentation is the backbone of scientific research (Karimzadeh & Hoffman, 2018; Witzman et al., 2020; Aghajani et al., 2020): if you write it clearly and release it online openly without restrictions, anyone who will reuse your software scripts will take advantage of it. On the contrary, the absence of well-written documentation would make it impossible for other people to use your scripts and programs. Examples of well-written tutorials for EEG software can be found on the already-mentioned MNE-Python package website (MNE, 2024). Moreover, keeping a detailed laboratory notebook is essential in this context (Schnell, 2015).

These practice for open scientific research, if taken into practice, will not only make your studies more robust and reliable, but will also boost your career in several ways, giving you more visibility (Fig. 1).

Conclusions

Understanding how the human brain works has always been a fascinating and difficult task for scientific researchers, and electroencephalography (EEG) has been a useful tool for this scope. Even if collecting EEG signal is a non-trivial task, analyzing electroencephalographic data remains one of the most informative way for investigating the dynamic functioning of human brain.

EEG is, indeed, employed in many different clinical neuroscience applications, from diagnosis of epilepsy to motor rehabilitation and EEG-driven wheelchairs. A similar variety is also present in the equipment that can record EEG: some are specifically dedicated to high-level research (with lots of electrodes, implying long preparation times but also high-quality data), others target portability and usability (being embedded with a few electrodes allocated in a more fancy and usable support, but providing much lower data quality). Thus, to face any type of data possibly acquired, a solid acquisition protocol and pre-processing pipeline are needed. Therefore, we presented here our ten quick tips for EEG signal acquisition and processing, by proposing some easy guidelines to avoid common mistakes in clinical neuroscientific research projects involving electroencephalographic data. Some of these quick recommendations derive from our direct experience, while others come from errors we noticed in other people’s studies or in published articles on EEG acquisition and/or analysis. We presented our ten pieces of advice in a simple way, so that they could be understood by anyone, including beginners and unexperienced researchers. Although we originally designed these tips for novices, we believe our ten quick tips should be followed and kept in mind by experienced researchers as well.

List of abbreviations

AEP auditory evoked potential

BCI Brain-Computer Interface

CAR common average

CC Creative Commons

CRAN Comprehensive R Archive Network

CT computed tomography

ECG electrocardiogram

EEG electroencephalography

EMG electromyograph

ERD event-related desynchronization

fMRI functional magnetic resonance imaging

Hz Hertz

ICA independent component analysis

iEEG intracranial electroencephalography

IRB Institutional Review Board

LORETA low-resolution electromagnetic tomography algorithm

ML machine learning

MUSIC multiple signal classification

NIRS near-infrared spectroscopy

p-value probability value

SSVEP steady-state visual evoked potentials

WMN weighted minimum-norm least squares

Additional Information and Declarations

Competing Interests

Author Contributions

Data Availability

Both Davide Chicco and Giulia Cisotto are academic editors at PeerJ Computer Science.

Giulia Cisotto conceived and designed the experiments, performed the experiments, analyzed the data, performed the computation work, prepared figures and/or tables, authored or reviewed drafts of the article, recommendations design and search in the literature, and approved the final draft.

Davide Chicco conceived and designed the experiments, performed the experiments, analyzed the data, performed the computation work, prepared figures and/or tables, authored or reviewed drafts of the article, recommendations design and search in the literature, and approved the final draft.

The following information was supplied regarding data availability:

There is no raw data or code in this study because it is a literature review.

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
