# Peer review of "Ten quick tips for clinical electroencephalographic (EEG) data acquisition and signal processing"

_PeerJ Computer Science, doi:10.7717/peerj-cs.2256_

## Round 0.1 · original submission · Minor Revisions

Please find attached the reviewer comments. We kindly request that you carefully revise your submission based on these valuable insights. Should you have any questions or require further clarification, please do not hesitate to reach out to us. We look forward to receiving the revised version at your earliest convenience.

Reviewer 1 ·

Basic reporting

The authors have presented ten quick tips for EEG data acquisition and processing. They have tried to highlight the basics of what needs to be done to get a novice researcher to get started in the field, while also providing guidance to experts so that they can avoid common mistakes. The paper, though is very basic, holds importance to novice researchers.

Experimental design

Below are a few suggestions/recommendations to improve the paper:
1. It would be good to mention various applications of EEG signal processing and provide references so that the readers get an insight of the different areas EEG signal processing is applied. A few examples would be emotion recognition [1, 2], sleep stage classification [3, 4], motor imagery classification [5-7], neurorehabilitation, seizure detection[8, 9], etc. It would also be good to mention commonly used methods or features for EEG signal analysis for the betterment of the readers. Some of the references that may be included are listed below.

1. Yao, X., et al., Emotion Classification Based on Transformer and CNN for EEG Spatial–Temporal Feature Learning. 2024. 14(3): p. 268.
2. Xu, G., W. Guo, and Y. Wang, Subject-independent EEG emotion recognition with hybrid spatio-temporal GRU-Conv architecture. Medical & Biological Engineering & Computing, 2023. 61(1): p. 61-73.
3. Zaman, A., et al., SleepBoost: a multi-level tree-based ensemble model for automatic sleep stage classification. Medical & Biological Engineering & Computing, 2024.
4. Masad, I.S., A. Alqudah, and S. Qazan, Automatic classification of sleep stages using EEG signals and convolutional neural networks. PLoS One, 2024. 19(1): p. e0297582.
5. Miah, M.O., et al., CluSem: Accurate clustering-based ensemble method to predict motor imagery tasks from multi-channel EEG data. Journal of Neuroscience Methods, 2021. 364.
6. Kumar, S., T. Tsunoda, and A. Sharma, SPECTRA: a tool for enhanced brain wave signal recognition. BMC bioinformatics, 2021. 22(6): p. 1-20.
7. Kumar, S., A. Sharma, and T. Tsunoda, Subject-Specific-Frequency-Band for Motor Imagery EEG Signal Recognition Based on Common Spatial Spectral Pattern, in 16th Pacific Rim International Conference on Artificial Intelligence (PRICAI 2019). 2019, Springer-Verlag: Cuvu, Yanuka Island, Fiji. p. 712–722.
8. Shen, M., et al., A real-time epilepsy seizure detection approach based on EEG using short-time Fourier transform and Google-Net convolutional neural network. Heliyon, 2024. 10(11): p. e31827.
9. Sukaria, W., et al., Epileptic Seizure Detection Using Convolution Neural Networks, in 2022 IEEE International Symposium on Medical Measurements and Applications (MeMeA). 2022. p. 1-5.

2. The English language should be improved. For example, the last sentence in the introduction section, could be rephrased as “….algorithms have already been proposed for EEG signal analysis.”. Line 61, “explained, and clearly stated.”. Line 84, “…team can start to recruit patients …”. Line 89, double dot at the end. Line 101, “medical equipe has” should read “medical equipment has”. Line 209, the word “performed” to be used instead of “operated”. Likewise, the whole manuscript contains errors and needs proofreading.
3. It would be good if the authors could also mention that the environment in which the different methods are evaluated needs to be the same to make a fair comparison between various techniques/methods, such as the same dataset, the same preprocessing methods, the same epoch size, etc. It has been noted that this is a common mistake made in this research domain.

Validity of the findings

The tips highlighted are important for novice researchers.

Reviewer 2 ·

Basic reporting

This well-written manuscript describes the use of electroencephalography (EEG) in recording brain activity and the challenges associated with processing and analyzing EEG data using computational methods. It emphasizes the need for careful analysis to avoid errors that could lead to inaccurate or misleading scientific conclusions in clinical neuroscience research. With some modifications, this manuscript will be very useful for researchers who attempt to work with EEG. My suggestions are as follows:

1. The tip numbers in Figure 1 are not consistent with the tip number in the content; for example, “Ethical Approval” is “Tip 2” in the figure, but in the content, it is listed as “Tip 1.” This can be confusing for the readers. Please modify the orders or use other terms.
2. Figure 1: The items listed in the box of tip 1 need to be clarified. For example, “neurology” is a branch of medicine, whereas “meditation” is a practice or technique.
3. Ln 78: “If working at the hospital, the research protocol has to be approved before it can be implemented.” The statement is inaccurate. Any research involving humans must be approved by the IRB and informed consent from the subjects before participation. Approval is needed for human studies conducted on campus, in the neighborhoods, etc, even with healthy subjects. Please revise this statement to avoid misleading.
4. Tip 2:
(1) I suggest adding some reminders regarding the settings, including recording environments (shielding room?), chair material, time of the day, and so on.
(2) Ln 111-112, “This might not hold true if…” This sentence is missing a noun and is unclear to me.
(3) Ln 138, I suggest providing an example of the occipital alpha rhythm so readers can catch it more easily.
5. Authors may consider using bullet points to ensure all tips are consistent.
6. Tip 4:
(1) “subject”-specificity and intra- “individual”; please unify the terms.
(2) Ln 203-204, please check grammar.
7. Ln 252, please check if it is weighted minimum norm (WMN) or weighted least square.
8. Ln 360, in addition to emphasizing publication in open-access journals, the authors could add a description on avoiding predatory journals.
9. Missing “AEP” and “iEEG” in the “List of abbreviations.”

Experimental design

no comment

Validity of the findings

no comment

Additional comments

no comment

---

## Round 0.2 · accepted · Accept

After careful assessment of the revised manuscript alongside a previous reviewer, we are pleased to confirm that we believe it is now ready for publication. We appreciate the efforts made in addressing the feedback provided.

Reviewer 2 ·

Basic reporting

I have reviewed the responses provided by the authors to my previous comments. Their responses have satisfactorily addressed all my concerns. I have no further questions or comments at this time. Thank you for the opportunity to review this manuscript.

Experimental design

Please see 1. Basic reporting.

Validity of the findings

Please see 1. Basic reporting.

Additional comments

Please see 1. Basic reporting.